# Evaluation of Small Molecule Combinations against Respiratory Syncytial Virus In Vitro

**DOI:** 10.3390/molecules26092607

**Published:** 2021-04-29

**Authors:** Yuzhen Gao, Jingjing Cao, Pan Xing, Ralf Altmeyer, Youming Zhang

**Affiliations:** State Key Laboratory of Microbial Technology, Microbial Technology Institute, Shandong University, Qingdao 266237, China; 201520230@mail.sdu.edu.cn (Y.G.); caojingjing@sdu.edu.cn (J.C.); xingpan@sdu.edu.cn (P.X.)

**Keywords:** RSV, fusion inhibitors, RdRp inhibitors, combination therapy, synergistic enhancement

## Abstract

Respiratory syncytial virus (RSV) is a major pathogen that causes severe lower respiratory tract infection in infants, the elderly and the immunocompromised worldwide. At present no approved specific drugs or vaccines are available to treat this pathogen. Recently, several promising candidates targeting RSV entry and multiplication steps are under investigation. However, it is possible to lead to drug resistance under the long-term treatment. Therapeutic combinations constitute an alternative to prevent resistance and reduce antiviral doses. Therefore, we tested in vitro two-drug combinations of fusion inhibitors (GS5806, Ziresovir and BMS433771) and RNA-dependent RNA polymerase complex (RdRp) inhibitors (ALS8176, RSV604, and Cyclopamine). The statistical program MacSynergy II was employed to determine synergism, additivity or antagonism between drugs. From the result, we found that combinations of ALS8176 and Ziresovir or GS5806 exhibit additive effects against RSV in vitro, with interaction volume of 50 µM^2^% and 31 µM^2^% at 95% confidence interval, respectively. On the other hand, all combinations between fusion inhibitors showed antagonistic effects against RSV in vitro, with volume of antagonism ranging from −50 µM^2^ % to −176 µM^2^ % at 95% confidence interval. Over all, our results suggest the potentially therapeutic combinations in combating RSV in vitro could be considered for further animal and clinical evaluations.

## 1. Introduction

Human respiratory syncytial virus (RSV) is the foremost cause of pneumonia and bronchiolitis in infants and young children, causing 3.4 million new pediatric cases of distal airway disease, and ~200,000 deaths every year worldwide [1,2]. It is also a significant cause of morbidity in the elderly [3], the immunocompromised [4] and transplant recipients [5]. Despite unremitting efforts, no effective and specific RSV treatment or vaccine is available to date [6,7], and the approved therapeutic options are ribavirin and monoclonal antibody palivizumab, which are inadequate due to severe side effects and exorbitant cost, respectively [8]. This situation produces an urgent need for discovering effective therapeutics for fighting RSV infection.

RSV is a member of the species Human orthopneumovirus and belongs to the Pneumoviridae genus of the family Pneumoviridae, order Mononegavirales [9]. The 15.2 kb non-segmented genome contains 10 mRNA species encoding 11 distinct proteins, including three surface glycoproteins (SH, G, and F) and several proteins (N, P, L, and M2-1) that comprise the viral RNA-dependent RNA polymerase complex (RdRp) [10]. The RdRp complex is capable of transcribing sub-genomic capped and poly-adenylated mRNAs and replicating the genome [11]. The F protein is critical for the fusion of the viral membrane with the cell plasma membrane [12], leading to infection of the host cells [13]. According to their significant functions in the RSV life cycle, fusion protein and RdRp complexes have become key targets of drug development [14,15,16].

In recent years, numerous small molecule compounds have been discovered to have the ability to block RdRp complex or to inhibit RSV fusion in vitro [17,18], for example, GS5806, JNJ-53718678, Ziresovir, ALS-008176, Sisunatovir and PC786. GS5806, Ziresovir and BMS433771 have been shown to be fusion inhibitors. GS5806 is an oral inhibitor that blocks pre-fusion protein, and Ziresovir is an RSV fusion inhibitor that is demonstrated to reduce cytopathic effects in infected HEp-2 cells. BMS433771 binds to pre-fusion F protein and stabilizes its conformation. In addition, ALS8176, RSV604 and CPM have been shown to be RdRp complex inhibitors and target N, L and M2-1, respectively. ALS8176 inhibits RSV polymerase activity, and RSV604 binds to N protein. In 2015, we first identified steroidal alkaloid Cyclopamine (CPM) [19], an antagonist of hedgehog (Hh)-signaling pathway, as an effective inhibitor of RSV replication. CPM inhibits RSV through a Smo-independent mechanism that targets M2-1 protein, and the resistance mutation M2-1 (R151K) is found at a domain binding to RNA and P protein. RSV604 (Astra Zeneca) is a nucleoprotein (N) inhibitor of RSV [20], and the resistance mutations selected in vitro are N(I129L) and (L139I), for which a finished Phase 1 clinical study has already shown their safety and tolerance in healthy adults [21]. ALS8176 (Alios Janssen) is a cytidine nucleoside analog blocking RdRp complex [22], for which four resistant mutations, L(M628L), L(A789V), L(L795I) and L(I796V), were identified in vitro. At present, clinical trials in RSV-infected hospitalized infants are ongoing [22]. As for fusion inhibitors, the first orally available inhibitor GS5806 (Gilead Sciences) [23], which blocks viral entry, has now finished a phase 2 clinical trial [24]. Ziresovir (Ark Biosciences) is another promising RSV fusion inhibitor and is currently in phase 3 clinical trials (NCT04231968) [25]. BMS433771 [26] is a RSV inhibitor of F protein that induces membrane fusion in vitro. Several resistant mutations are located in the F1 domain, which is orally available efficacious in rodent models [27].

A therapeutic combination strategy for RSV treatment might be an attractive option to overcome drug-associated side effects and antiviral resistance development [28,29]. Similar combination strategies are effective against HCV [30], HCMV [31,32], EV71 [33], influenza [34,35], and ZIKV [36]. As a proof-of-concept of combination therapeutic strategies against RSV, two-compound combinations of fusion inhibitors (GS5806, Ziresovir, BMS433771) and RdRp inhibitors (ALS8176, RSV604, CPM) were evaluated in HEp-2 cells with a firefly luciferase gene inserted into a recombinant RSV strain as described recently, and inhibition efficacy of antiviral combinations were measured by luciferase activity and quantitative PCR.

## 2. Results

### 2.1. Antiviral Activity of Individual Compounds against RSV In Vitro

To evaluate the potency of six inhibitors (Figure 1), the antiviral activities were determined by luciferase activity reduction assay as described previously [9]. Meanwhile, cytotoxicity assays were done with the Cell Titer-Glo Luminescent cell viability assay (Promega) after incubation with the compounds to detect cell viability. IC50 was graphically defined by a 50% reduction in luciferase activity with Graphpad software (Graphpad Prism 8.0). As shown in Figure 2, IC_50_ and CC_50_ values of all single inhibitors were not statistically different from those published [37,38,39,40,41,42]. Cell viability was >90% in all combinations of compound concentrations (see Appendix A).

### 2.2. Antiviral Activity of GS5806 Combined with RdRp Inhibitors against RSV In Vitro

GS5806 has been previously demonstrated as a fusion inhibitor of RSV. We, therefore, measured the interactions between GS5806 and three RdRp inhibitors ALS8176, RSV604 and CPM. For the combination of GS5806 with ALS8176, all values on the Z-axis in graph were above the additive plane (Figure 3a). The results showed additive effects against RSV in vitro via luciferase activity reduction assay, the volume of interaction was 31.53 µM^2^% at 95% confidence interval. In addition to this method, we validated the interaction between them using a qPCR-based method that differs from luciferase activity reduction assay with the same assay format (Figure 3d). The result showed slight synergy between GS5806 and ALS8176, with the interaction volume of 58 µM^2^% at 95% confidence interval. Using these two different detection methods, we conclude that the combination of GS5806 and ALS8176 showed an additive effect against RSV in vitro.

The GS5806–RSV604 combination produced a complex interaction profile consistent with the curves presented (Figure 3b), with interaction volume great than zero at low concentrations (<1.5 nM for GS5806 and <1.5 µM for RSV 604) of both agents and interaction volume less than zero at high concentrations, and the volume of interaction was 31 µM^2^%/−5 µM^2^% at 95% confidence interval, this combination showed an additive effect in vitro. The combination of GS5806 with CPM resulted in an additive effect (Figure 3c), the volume of interaction was −19 µM^2^% at a 95% confidence interval.

### 2.3. Antiviral Activity of Ziresovir Ccombined with RdRp Inhibitors against RSV In Vitro

Ziresovir is another promising fusion inhibitor of RSV that is currently in phase 3 clinical trials; therefore, we measured the interactions between Ziresovir in combination with three RdRp inhibitors: ALS8176, RSV604 and CPM. For the combination of Ziresovir with ALS8176, all values on the Z-axis in the graph were above the additive plane (Figure 4a), the volume of synergy was 50 µM^2^% at 95% confidence interval, we conclude that Ziresovir–ALS8176 showed an additive effect against RSV in vitro.

In addition to ALS8176, we measured the interaction between Ziresovir and the other two RdRp inhibitors: RSV604 and CPM. For the combinations of RSV604 and CPM with Ziresovir, all of the values on the Z-axis in the graph were at or below the additive plane (Figure 4b,c), the volumes of interaction were −5 µM^2^% and −25 µM^2^% at 95% confidence interval, respectively. Ziresovir–RSV604 and Ziresovir–CPM resulted in additive effects against RSV in vitro.

### 2.4. Antiviral Activity of BMS433771 Combined with RdRp Inhibitors against RSV In Vitro

Using the same assay as above, we evaluated the combinations of BMS433771, another fusion inhibitor, with three RdRp inhibitors (ALS8176, RSV604 and CPM). For the combination of BMS433771 with ALS8176, all the values on the Z-axis in the graph were above the additive plane (Figure 5a), the volume of interaction was 16 µM^2^% at 95% confidence interval. We conclude that BMS433771–ALS8176 resulted in an additive effect against RSV in vitro. As for the combination of BMS433771 with RSV604, almost all the values in the graph were above the additive plane (Figure 5b), except the point (BMS433771 0.3 nM, RSV604 0.05 μM). In addition, the volume of synergy was 64 µM^2^% at 95% confidence interval, this combination resulted in a slight synergy effect against RSV in vitro.

For the interaction between BMS433771 and CPM, all the values on the Z-axis in the graph were below the additive plane (Figure 5c), and the volume of interaction was −18 µM^2^% at 95% confidence interval. We conclude that BMS433771–CPM showed an additive effect against RSV in vitro.

### 2.5. Antiviral Activity of Two RdRp Inhibitors in Combination against RSV In Vitro

Although ALS8176, RSV604 and CPM had the same target (RSV RdRp complex), resistance profiles for the three inhibitors demonstrated that they targeted different components (ALS8176, L protein; RSV604, N protein; and CPM, M2-1 protein). We measured the interactions between two RdRp inhibitors using the same method as above. The ALS8176–RSV604 combination produced a complex interaction profile consistent with the curves presented (Figure 6a), with interaction volume great than zero at low concentrations (<0.3 µM for ALS8176 and <1.5 µM for RSV 604) of both agents and interaction volume less than zero at high concentrations, and the volume of synergy/antagonism was 20 µM^2^%/−5 µM^2^% at 95% confidence interval, we conclude that this combination showed an additive effect in vitro.

For the CPM–ALS8176 and CPM–RSV604 combinations (Figure 6 b,c), both combinations showed additive effects against RSV in vitro. The volumes of interaction were −20 µM^2^% and −50 µM^2^% at 95% confidence intervals, respectively.

### 2.6. Antiviral Activity of Two Fusion Inhibitors in Combination against RSV In Vitro

In this part, we measured the interactions between two fusion inhibitors using the same method as above. For the interaction between GS5806 and Ziresovir, all the values on the Z-axis in the graph were below the additive plane (Figure 7a), and the volume of interaction was −50 µM^2^% at 95% confidence interval. We conclude that GS5806—Ziresovir showed an additive effect against RSV in vitro. As for the combination of GS5806 with BMS433771, almost all the points in the graph were below the additive plane (Figure 7b), and the volume of interaction was −36 µM^2^% at 95% confidence interval. The GS5806–BMS433771 also showed an additive effect against RSV in vitro.

For the interaction between Ziresovir and BMS433771, all the values on the Z-axis in the graph were below the additive plane (Figure 7c), and the volume of antagonism was −176 µM^2^% at 95% confidence interval. We conclude that Ziresovir–BMS433771 showed a strong antagonism effect against RSV in vitro.

## 3. Discussion

Pharmacodynamic compound interactions are complex and not always predictable. They require detailed knowledge of the pharmacokinetic drug profiles when given in combination before clinical translation. In theory, synergy can be expected if the two compounds being investigated target different processes or competitively target distinct parts in the same processes; however, this is not always true, and thus in vitro studies are used to confirm these types of interactions. Transferring in vitro findings into clinical applications thus remains a challenge and might require the development of neotype evaluation methods and treatment approaches.

In this study, we selected six promising compounds based on their different mechanisms of action for combination studies. Among them, GS5806, Ziresovir andBMS433771 are fusion inhibitors, and ALS8176, RSV604, and CPM are RdRp inhibitors targeting different components. We designed two compound combination strategies, including fusion inhibitor–RdRp inhibitor, RdRp inhibitor–RdRp inhibitor and fusion inhibitor–fusion inhibitor. As noted in Table 1, all three fusion inhibitors showed additive enhancement of effects when combined with ALS8176 and RSV604 (<1.5 μM). Three combinations between fusion inhibitors showed strong antagonistic effects against RSV in vitro. Synergy can be expected if the two compounds being investigated target different steps of virus replication [31]. Previous studies indicated that GS5806, BMS433771 and Ziresovir interfere with RSV F protein through a same mode of inhibition, targeting the same domain (between HR1 and HR2) [6]. It is likely that the molecules inhibitors compete with each other when interaction with F protein, resulting in the observed antagonism effect [31,33]. Interestingly, the combination of cyclopamine with any other compound yielded no more additive antiviral effects, and CPM–RSV604 showed a strong antagonistic effect against RSV in vitro. Since exact molecular mechanism of CPM is not fully understood, we guess it is the undesired off-target effects of CPM as a HH signal pathway inhibitor. ALS8176–RSV604 showed a synergistic enhancement of effect, consistent with what was reported previously [42].

As combination therapy becoming the standard treatment of viral infections, interactions between antivirals should be tested to select combinations which offer a synergistic enhancement of effect [34]. This study demonstrated the initial proof-of-concept of a direct-acting antiviral combination strategy against RSV in in vitro assays. The selected fusion inhibitors showed synergistic enhancement of effect when combined with promising RdRp inhibitors. We consider combining inhibitors which targeting different steps of RSV replication, to be most promising, since pan-resistance to entry inhibitors is already described [8]. Further assessment of these combinations in vivo might pave a way for the development of effective RSV therapeutic strategies.

## 4. Materials and Methods

### 4.1. Chemicals

CPM (S1146) was purchased from Selleck Chemicals. RSV604 (SML1695) and BMS433771 (BMS0022) were purchased from Sigma–Aldrich. ALS8176 (HY–12983A), GS5806 (HY–16727), and Ziresovir (HY–109142) were purchased from MedChem Express. Compounds were solubilized at 8 mM in 100% *v/v* DMSO, then serially diluted to the desired concentrations in DMEM with 2% fetal bovine serum (FBS) and 0.25% *v/v* DMSO in the assay.

### 4.2. Cell and Virus

HEp-2 cells (ATCC reference CCL-23) were grown in DMEM supplemented with 10% (*v/v*) FBS and penicillin–streptomycin. The cells were grown in an incubator at 37 °C in 5% CO_2_. The RSV–Luc strain engineered from RSV Long strain (ATCC reference VR–26) was a gift from Dr. Jean-François Eleouet, INRA, France.

### 4.3. Virus Iinhibition and Cytotoxicity Assays

HEp-2 cells were added into a 96-well plate with 5 × 10^4^ cells per well one day prior to being infected with 3000 plaque-forming units (PFU) of RSV–Luc in the presence of various compounds or combined compounds concentrations. Before being added to each plate, all set compounds/compound dilutions were incubated at 37 °C with viral suspension in an incubator for 5 min. For combined treatment, both single compound controls were present on each plate, which is necessary for the results to be valid. Plates were incubated at 37 °C for 48 h, then washed with PBS before lysis with 50 μL of lysis buffer (25 mM Tris pH 7.8, 8 mM MgCl_2_, 15% glycerol, 1% Triton X-100) per well for 15 min at room temperature with gentle shaking. Plate and substrate (Bright-Glo™ Luciferase Assay System; Promega) were equilibrated at 37 °C for 10 min before luciferase activity measurement, and luciferase activity was read using Biotek Synergy H1microplate reader. Relative luciferase activity refers to the percentage of luciferase activity relative to the mean value of 6 infected wells treated with 0.25% *v/v* DMSO. Different inhibitions were calculated by fitting to the sigmoidal curve equation (Graphpad software 8.0). Cytotoxicity was conducted in the same way as antiviral activity but without viral infection, and cytotoxicity was determined by the CellTiter–Glo Luminescent cell viability assay (Promega).

### 4.4. Quantitative PCR

Hep-2 cells were added into a 48-well plate with 1 × 10^5^ cells one day prior to being infected with 6000 PFU of RSV–Luc in the presence of various compounds or combined compound concentrations. Before being added to each plate, all set compound/compounds dilutions were incubated with viral suspension in an incubator at 37 °C for 5 min. Plates were incubated at 37 °C for 48 h, and total RNA in cells was extracted using the QIAamp viral RNA minikit (Qiagen). RNA was reverse-transcribed using the cDNA reverse-transcription kit (Thermo) with random primers.

Quantitative real-time PCR (qRT–PCR) analysis was performed to amplify SH–G (F: TGCAAACCACCATCCATA; R: CCTAGTTCATTGTTATGA) intergenic region using the cDNA as the template and GAPDH (F: CCATGTTCGTCATGGGTGTGAACCA; R: GCCAGTAGAGGCAGGGATGATGTTC) cDNA as the internal standard. The relative number of viral RNA copies was calculated using the 2−ΔΔCt method. Each experiment was repeated in triplicate, and different inhibitions were calculated by fitting to the sigmoidal curve equation (Graphpad software 8.0).

### 4.5. Data Analysis

Two single compounds with similar effects sometimes produce an impaired or exaggerated consequence when used in combination. To detect the compound interaction, the inhibition result from luciferase activity reading and viral genome was analyzed with MacSynergy II as described previously [37]. This program employs the Bliss Independence algorithm for calculating drug combination interaction to derive the volume of the peaks at different drug combinations. This model calculates a theoretical additive interaction from the dose–response curves of each compound used. The calculated additive surface would appear as a horizontal plane at 0, peaks above this plane indicate synergism, and depressions below this plane indicate antagonism. In this study, the volumes (μM^2^%) at 95% confidence interval of the peak (or depression) were calculated, they represent the lower value of this interval for positive values and the higher value of this interval for negative volumes and were defined as follows: The volumes of greater than +100 are considered as strong synergy, volumes between +50 and +100 are considered as slight synergy, values between −50 to +50 are considered additive. Similarly, values between −100 and −50 are considered as slight antagonism, values of less than −100 are strong antagonism [43].

## Figures and Tables

**Figure 1 molecules-26-02607-f001:**
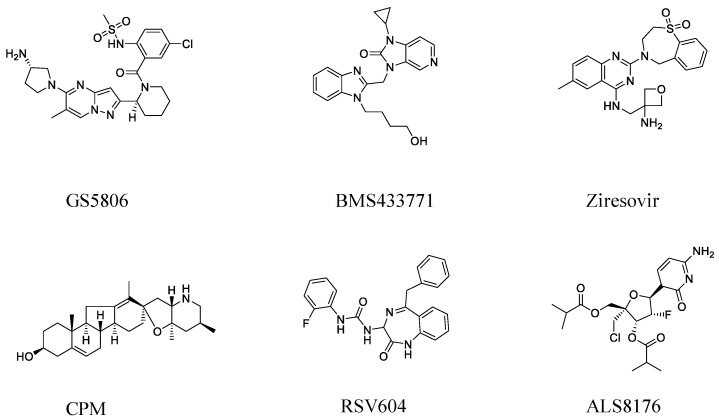
The chemical structural formulas of GS5806, BMS433771, Ziresovir, CPM, RSV604, and ALS8176.

**Figure 2 molecules-26-02607-f002:**
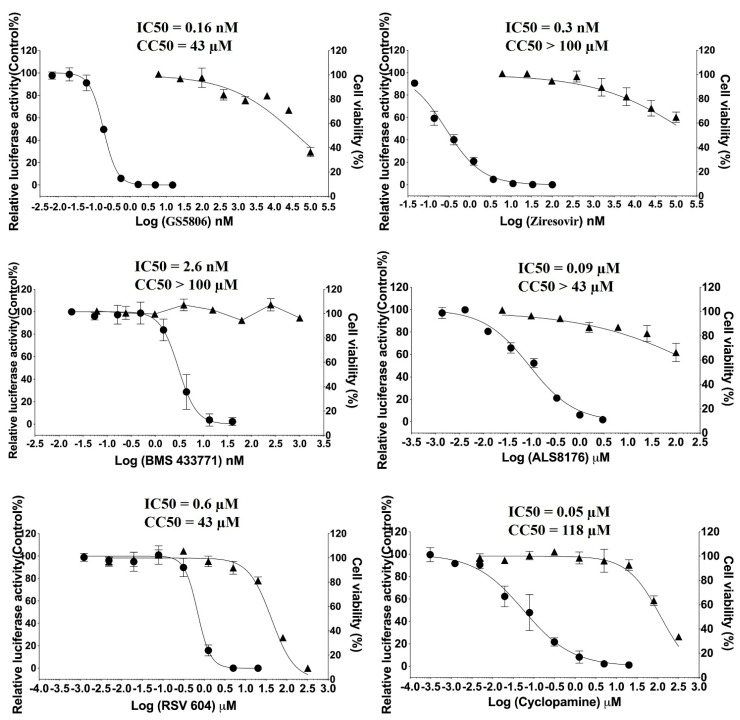
Antiviral activity of selected inhibitors. HEp-2 cells in 96-well plates were infected with respiratory syncytial virus (RSV)–Luc strain in the presence of various dilutions of selected compounds or DMSO control. Luciferase activity (•) was measured, normalized, and expressed as % of DMSO control. The cytotoxicity (▲) of compounds was measured in parallel in the same cells. The results are representative of three independent experiments performed in triplicate (mean ± s.d.).

**Figure 3 molecules-26-02607-f003:**
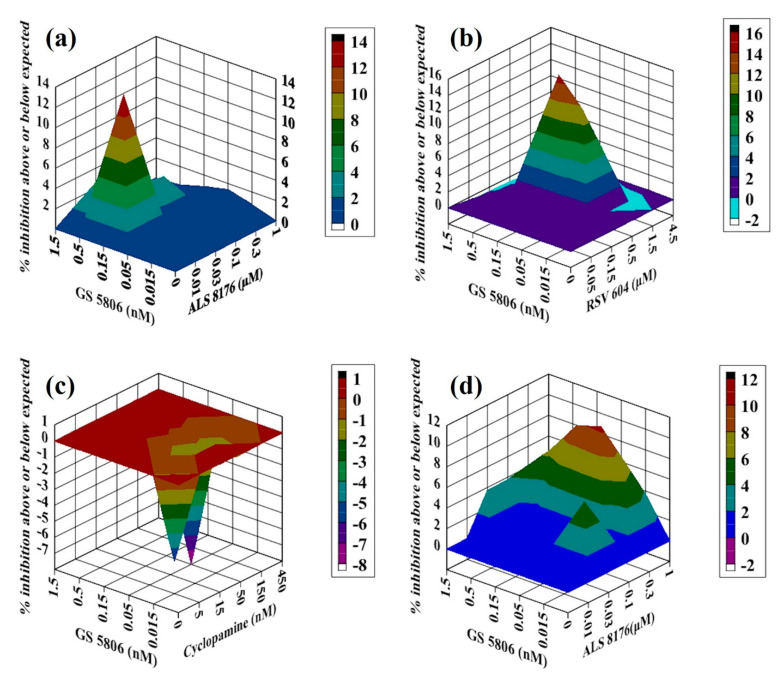
Analysis of compound interactions between RSV fusion inhibitor GS5806 and RdRp inhibitors in vitro. HEp-2 cells infected with RSV-Luc were subjected to serial dilutions of GS5806 ((**a**–**d**); 0.015–1.5 nM), ALS8176 ((**a**,**d**); 0.01–1 μM), RSV604 (**b**); 0.05–4.5 μM), or CPM (**c**); 5–450 nM) in a checkerboard method, and results were obtained via a luciferase activity reduction assay (**a**–**c**) or qPCR (**d**). Data shown were obtained using Macsynergy II program at 95% confidence interval and plotted with Delta Graph, and values on the *Z*-axis represent the mode of interaction between two selected compounds (>0, synergistic effect; <0, antagonistic effect; 0, additive effect). All data points represent averages of three measurements from three experiments.

**Figure 4 molecules-26-02607-f004:**
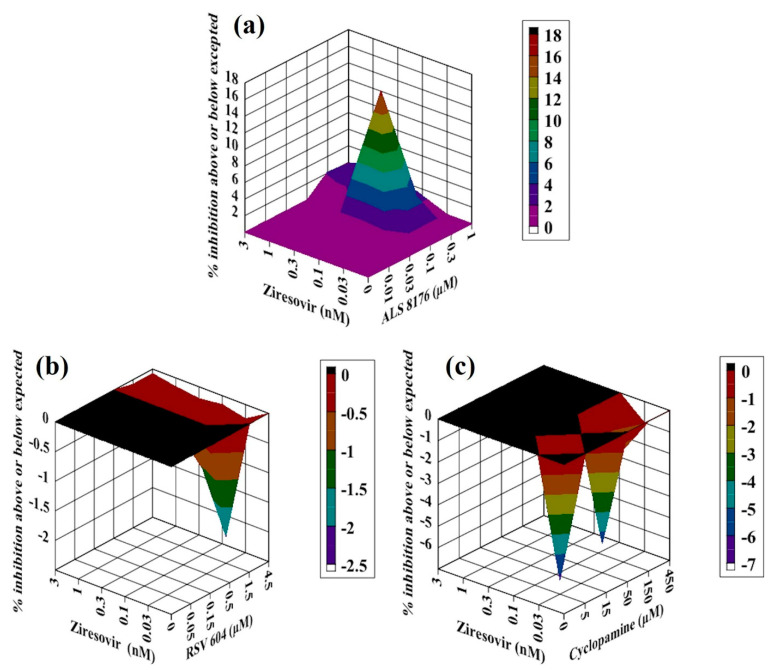
Analysis of compound interactions between RSV fusion inhibitor Ziresovir and RdRp inhibitors. HEp-2 cells infected with RSV–Luc were subjected to serial dilutions of Ziresovir ((**a**–**c**); 0.03–3 nM), ALS8176 ((**a**); 0.01–1 μM), RSV604 ((**b**); 0.05–4.5 μM), or CPM ((**c**); 5–450 nM) in a checkerboard method, and results were obtained via a luciferase activity reduction assay (**a**–**c**). Data shown were obtained using Macsynergy II program at 95% confidence interval and plotted with Delta Graph, and values on the Z-axis represent the mode of interaction between two selected compounds (>0, synergistic effect; <0, antagonistic effect; 0, additive effect). All data points represent averages of three measurements from three experiments.

**Figure 5 molecules-26-02607-f005:**
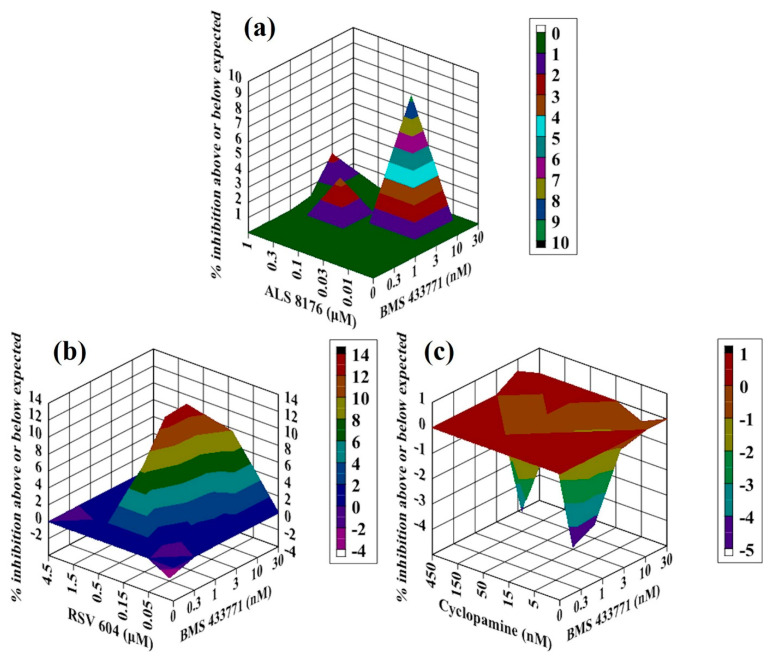
Analysis of compound interactions between RSV fusion inhibitor BMS433771 and RdRp inhibitors. HEp-2 cells infected with RSV–Luc were subjected to serial dilutions of BMS433771 ((**a**–**c**); 0.3–30 nM), ALS8176 ((**a**); 0.01–1 μM), RSV604 ((**b**); 0.05–4.5 μM), or CPM ((**c**); 5–450 nM) in a checkerboard method, and results were obtained via a luciferase activity reduction assay (**a**–**c**). Data shown were obtained using Macsynergy II program at 95% confidence interval and plotted with Delta Graph, and values on the Z-axis represent the mode of interaction between two selected compounds (>0, synergistic effect; <0, antagonistic effect; 0, additive effect). All data points represent averages of three measurements from three experiments.

**Figure 6 molecules-26-02607-f006:**
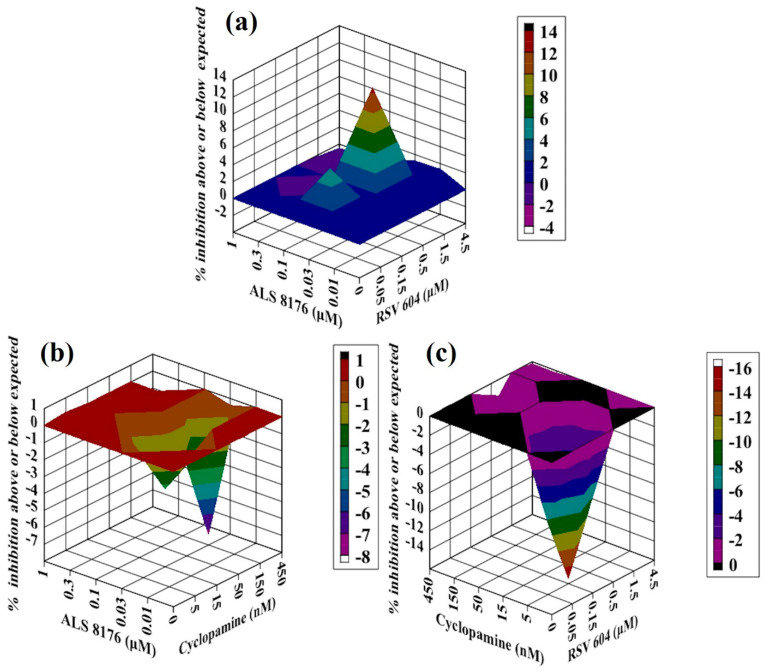
Analysis of compound interactions between RdRp inhibitors. HEp-2 cells infected with RSV–Luc were subjected to serial dilutions of ALS8176 ((**a**,**b**); 0.3–30 nM), RSV604 ((**b**,**c**); 0.05–4.5 μM), or CPM ((**a**,**c**); 5–450 nM) in a checkerboard method, and results were obtained via a luciferase activity reduction assay (**a**–**c**). Data shown were obtained using Macsynergy II program at 95% confidence level and plotted with Delta Graph, and values on the Z-axis represent the mode of interaction between two selected compounds (>0, synergistic effect; <0, antagonistic effect; 0, additive effect). All data points represent averages of three measurements from three experiments.

**Figure 7 molecules-26-02607-f007:**
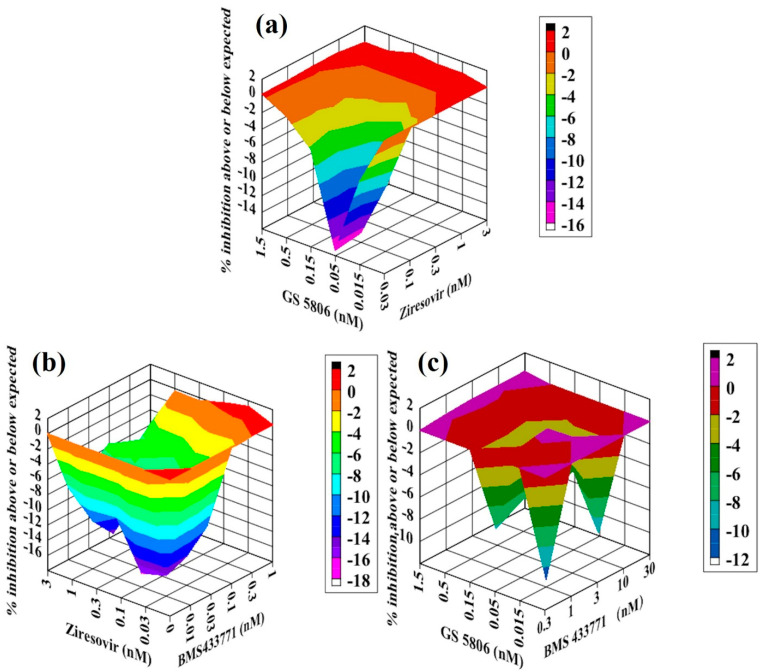
Analysis of compound interactions between fusion inhibitors. HEp-2 cells infected with RSV–Luc were subjected to serial dilutions of GS5806 ((**a**,**b**); 0.015–1.5 nM), Ziresovir ((**a**,**c**); 0.03–3 nM), or BMS433771 ((**b**,**c**); 0.3–30 nM) in a checkerboard method, and results were obtained via a luciferase activity reduction assay (**a**–**c**). Data shown were obtained using Macsynergy II program at 95% confidence interval and plotted with Delta Graph, and values on the Z-axis represent the mode of interaction between two selected compounds (>0, synergistic effect; <0, antagonistic effect; 0, additive effect). All data points represent averages of three measurements from three experiments.

**Table 1 molecules-26-02607-t001:** Interactions of compound–compound combinations against RSV.

Compound A	Compound B	Synergy/Antagonism (μM^2^%) ^a^ by MacSynergy II Analysis	Synergy/Antagonism ^a^
GS5806 ^c^	ALS8176 ^d^	31.53/0	Additivity
	ALS8176 ^b,d^	58/0	Slight synergy
	RSV604 ^d^	31/−5	Additivity
	Cyclopamine ^d^	0/−19	Additivity
	Ziresovir ^c^	0/−50	Additivity
	BMS433771 ^c^	0/−55	Slight antagonism
Ziresovir ^c^	ALS8176 ^d^	50/0	Additivity
	RSV604 ^d^	0/−5	Additivity
	Cyclopamine ^d^	0/−25	Additivity
	BMS433771 ^c^	0/−176	Strong antagonism
BMS433771 ^c^	ALS8176 ^d^	16/0	Additivity
	RSV604 ^d^	64/0	Slight synergy
	Cyclopamine ^d^	0/−18	Additivity
ALS8176 ^d^	RSV604 ^d^	20/−5	Additivity
	Cyclopamine ^d^	0/−20	Additivity
RSV604 ^d^	Cyclopamine ^d^	0/−50	Additivity

^a^. Mean volumes of synergy or antagonism are presented based on 95% confidence interval. ^b^. Values determined via qPCR. ^c^. fusion inhibor. ^d^. RdRp inhibitor.

## Data Availability

Not applicable.

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
