# Peer review of "Evaluation of Small Molecule Combinations against Respiratory Syncytial Virus In Vitro"

_molecules, 2021, doi:10.3390/molecules26092607_

Round 1
Reviewer 1 Report
Experiments well performed and explained in detail, effect of compounds and their combinations on antagonistic/synergistic activity against RSV replication exmined , also by software, in a cellular viral replication model.
Makes sense as combinations of antivirals is very often common used and important to know that they have no antagonistic behavior.
Discussion is a bit short and somewhat more explanations would improve the value of the manuscript, why for example all fusion inhibitors are antagonistic and in other combinations not.
Reviewer 2 Report
The authors addressed the research question if a combination strategy with inhibitors with different mode of actions could be effective against RSV infection. As RSV infection is widely distributed and causes pneumonia and bronchiolitis worldwide, effective therapeutics are needed for fighting RSV infection. They therefore combined fusion inhibitors with RdRp inhibitors and evaluated their efficacy using a cell based assay.
Page 2, line 67: …as descriebed previously…please add a reference
The authors used a method from a commercial supplier for the assessment of cytotoxicity of the selected compounds. The combination of drugs was not assessed regarding their cytotoxicity. They used a luciferase based assay for detection of antiviral activity using Hep-2 cells. In the next step, they combined inhibitors and assessed the antiviral activity of these combinations using the same methods as before. As I am not used to data shown in isobolograms and have no experience with that, I am probably not the right person to say if the method used is the most current available and if it was used correctly. The quality of the experiments seems to be okay (technical replicates, biological replicates, statistical analysis)
Figure 3: The solution of the figure is really bad and I cannot read the text on the axis. I also have difficulties to understand the message from that figure.
Q: Why did the authors not assessed the combination of drugs for their cytotoxicity?
Three combinations between fusion inhibitors showed strong antagonistic effect against RSV in vitro. It demonstrated the initial proof-of-concept of a antiviral strategy against RSV in vitro. I could image that other researchers might be interested in reading the study. I completely missed an outlook for a further development of a combined therapy.
Q: Please add a bit more detailed outlook for further research.
The manuscript fit together very well. It is easy to read. It has some spelling etc. mistakes.
Author Response
Dear reviewer,
We appreciated for your efforts and found the comments were very useful. Please see the attachment.

Reviewer 3 Report
The authors explored the antiviral potency of several combinations of RSV antivirals. The results are most likely meaningful but there seems to be some misunderstanding of terms and some information cannot be read from the paper.
Major critique:
- In most figures the name of axis is in too small print to read. For this reason, we cannot understand what is shown.
- In fig 2 the circles and triangles are too small to distinquish so we cannot understand this figure.
- While the authors claim that clinical trials for several of the drugs are ongoing it has come to my attention that for most of these compounds this is not the case. If clinical trials for RSV inhibition are still ongoing, please refer to the clinical trial number. I believe only Ziresovir is still in active development.
- Since most compounds are not in development this exercise is rather a principle exercise and should be described in that way.
- The authors use the term “isobologram” to refer to some of the graphs. An isobologram is a graph where all the dots shown have the same % effect. This is not the case in the graphs depicted from MacSynergy.
- The authors refer to the term “Combination Index”. There is a specific definition for this term (see https://www.sciencedirect.com/science/article/abs/pii/S2213713018300051). The authors determine in fact the “Interaction Volume” as calculated using MacSynergy.
- The interaction volume has a unit being µM2 (as described in the methods). Please use this unit in the text.
- In table I one of the headers reads “95% Confidence Interval” but does not indicate what this is the interval from. This should be “Interaction Volume (95% Confidence Interval)”. Also indicate the unit (µM2).
- Same with “95%SynergyPlot”. What is this number? What is the unit?
- In line 70 the authors mention the lack of cytotoxicity of the combinations. But they do not indicate if the data is included in the paper, or not shown or what is the definition of lack of cytotoxicity in this case?
Author Response

(The authors gave the same response as above.)

Round 2
Reviewer 3 Report
In the introduction the authors describe several antivirals as "shown in vivo and clinical promise". To my knowledge the clinical development of all these molecules, except for Ziresovir, has been stopped. This should explicitly be stated. These molecules do not show "clinical promise". For example the authors write that clinical trials with ALS-8176 for RSV are ongoing but this is in my opinion not correct.
The authors currently do not indicate how the interaction volumes should be interpreted. As a comparison please check the paper: "Wildum S, Zimmermann H, Lischka P. 2015. In vitro drug combination studies of letermovir (AIC246, MK-8228) with approved anti-human cytomegalovirus (HCMV) and anti-HIV compounds in inhibition of HCMV and HIV replication. Antimicrob Agents Chemother 59:3140 –3148." Here the authors clearly stipulate volume limits that give a qualitative measure of the interaction volume: values of less than -100 reflect strong antagonism, values in the range -100 - -50 indicate low antagonism. Similarly values between 50 - 100 are considered low synergism, and values >100 indicate high synergy. This should then be phrased in this way in table 1. So not just "synergy" but "high synergy" or "low synergy". In my opinion all interaction volumes between 50 and -50 can be considered as "Additive".
Please also indicate in the methods what "x μM2% at 95% confidence interval." means. I believe it means that the software calculates the 95% confidence interval and returns the lower value of this interval for positive values and the higher value of this interval for negative volumes. Is this correct?
Perhaps also best to indicate in table 1 which compounds are fusion inhibitors and which are RDRP-inhibitors.